# Clinical Value of Galectin-9, Soluble TREM-1, and Soluble CD25 Among Critically Ill Patients with Organ Failure in the Emergency Department: A Prospective Observational Study

**DOI:** 10.3390/diagnostics15212677

**Published:** 2025-10-23

**Authors:** Uihwan Kim, Sijin Lee, Kap Su Han, Su Jin Kim, Sungwoo Lee, Dae Won Park, Juhyun Song

**Affiliations:** 1Department of Emergency Medicine, Korea University Anam Hospital, Seoul 02841, Republic of Korea; aniulpo2@korea.ac.kr (U.K.); reonoaz85@gmail.com (S.L.); hanks96@hanmail.net (K.S.H.); icarusksj@gmail.com (S.J.K.); kuedlee@korea.ac.kr (S.L.); 2Department of Internal Medicine, Korea University Ansan Hospital, Ansan-si 15355, Republic of Korea

**Keywords:** galectin-9, mortality, organ failure, sepsis, soluble CD25, soluble TREM-1

## Abstract

**Background/Objectives**: This study investigated clinical value of galectin-9 (Gal-9), a soluble triggering receptor expressed on myeloid cells-1 (sTREM-1), and soluble CD25 (sCD25) among critically ill patients with organ failure in the emergency department. **Methods**: Overall, 786 patients were enrolled and classified into non-infectious organ failure (NIOF, *n* = 331), sepsis (*n* = 266), and septic shock (*n* = 189). The diagnostic value of Gal-9, sTREM-1, and sCD25 were evaluated by receiver operating characteristic curve analysis. The prognostic value of the biomarkers was evaluated using Kaplan–Meier survival curve and Cox proportional hazard model analyses. **Results**: Gal-9, sTREM-1, and sCD25 could discriminate sepsis from NIOF (Gal-9, area under the curve [AUC], 0.599–0.678; sTREM-1, AUC, 0.616–0.695; sCD25, AUC, 0.710–0.781) and septic shock from sepsis (Gal-9, AUC, 0.562–0.667; sTREM-1, AUC, 0.572–0.676; sCD25, AUC, 0.555–0.660), respectively. Sepsis patients with higher levels of biomarkers over their cut-off value showed higher 30-day mortality compared to those with lower levels below the cut-off value (Gal-9 ≥ 14,391.80 ng/L, *p* < 0.001; sTREM-1 ≥ 580.62 ng/L, *p* < 0.001; sCD25 ≥ 1639.29 ng/L, *p* < 0.001; respectively) (log-rank test). sCD25 is an independent risk factor for 30-day mortality in patients with sepsis or septic shock. **Conclusions**: Gal-9, sTREM-1, and sCD25 showed diagnostic and prognostic value in critically ill patients with organ failure. sCD25 can predict the 30-day mortality in patients with sepsis. Gal-9, sTREM-1, and sCD25 could serve as auxiliary biomarkers to support clinicians in effective sepsis management.

## 1. Introduction

Sepsis is a life-threatening disease characterized by organ dysfunction, induced by a dysregulated host response to infection [1]. Although the surviving sepsis campaign (SSC) guidelines emphasize the importance of the early diagnosis of sepsis [2], there is no gold standard for the diagnosis of sepsis and no reliable tools to predict clinical outcomes.

Although blood cultures are useful to detect the presence of bacteremia, it requires a few days to obtain microbiological results, which might lead to false-negative outcomes. In clinical settings, biomarkers, such as C-reactive protein (CRP) and procalcitonin (PCT), have been widely used to identify and prognosticate sepsis patients [3]. Various other biomarkers, including CD64, presepsin, interleukin-6, and adrenomedullin have been evaluated and reported to provide valuable information for identifying sepsis and predicting clinical outcomes [4,5,6,7]. However, no single biomarker can perfectly reflect the pathological state of patients with sepsis, and they have several limitations in both diagnostic and prognostic value. Therefore, an investigation into the clinical value of novel biomarkers is required to establish a comprehensive care strategy for patients with sepsis.

Galectin-9 (Gal-9) is a potential organ dysfunction biomarker known for its immunomodulatory role in various microbial infections and is expressed in whole organ systems by mediating host-pathogen interactions [8]. During acute infection, it is assumed that Gal-9 is rapidly released as a danger signal to initiate innate immune cell activity [8]. Gal-9 reflects the severity of infectious diseases, such as malaria, dengue, tuberculosis, and leptospirosis [9,10,11,12]. However, to our knowledge, the clinical value of Gal-9 has not been assessed in patients diagnosed with sepsis in the emergency department (ED).

Triggering receptor expressed on myeloid cells-1 (TREM-1) is a member of the immunoglobulin superfamily that responds to the presence of bacterial components [13,14]. Amplification of the inflammatory response by TREM-1 is considered a critical contributor to the dysregulated immune response in sepsis. Pediatric patients admitted to intensive care units (ICUs) with septic shock have higher TREM-1 levels than those with severe sepsis [15]. Elevated levels of soluble TREM-1 (sTREM-1) are associated with increased mortality in patients with septic shock [16]. sTREM-1 levels are also increased in other body fluids such as urine and cerebrospinal fluid. While some studies reported good diagnostic and prognostic values of sTREM-1, other meta-analyses, including nine studies, reported only moderate sensitivity and specificity in predicting mortality [17].

CD25 is an interleukin (IL)-2 receptor that is expressed in correlation with regulatory FOXP3+ T cells, and soluble CD25 (sCD25), which is emitted into the blood and is measurable, reflects compensatory regulatory responses [18]. Immunosuppression in sepsis may be closely linked to the development of acute kidney injury (AKI), and sCD25 or IL-10 may be useful as novel biomarkers for the development of septic AKI [19]. Combined CD25, CD64, and CD69 biomarker panels can be used to effectively diagnose sepsis [20]. Huang et al. suggested that increased sCD25 levels correlated with poor clinical outcomes in patients [18].

Recent studies for the past three years have investigated the performance of Gal-9 for predicting the risk of malignancy in dermatomyositis [21], prognostic value of TREM-1 in sepsis [22], and diagnostic value of sCD25 for discriminating neonatal sepsis [23]. Although there have been several studies on the clinical value of sTREM-1 and sCD25, they have some limitations, such as study on non-infectious diseases, relatively small sample size, application of previous sepsis definitions, and ICUs-based study.

To the best of our knowledge, the clinical value of Gal-9 has not been assessed in critically ill patients with organ failure in the ED. Thus, we aimed to investigate the diagnostic and prognostic values of Gal-9, sTREM-1, and sCD25 in critically ill ED patients with non-infectious organ failure (NIOF), sepsis, and septic shock.

## 2. Materials and Methods

### 2.1. Study Design and Setting

This prospective observational study was conducted in the ED of a tertiary care teaching hospital. Our study adhered to the Declaration of Helsinki (2013; Seventh revision, 64th Meeting, Fortaleza) and was approved by the Institutional Review Board (IRB) of Korea University Ansan Hospital (IRB no. 2022AS0313). Verbal information about this study was provided to all study participants or their legal representatives in advance and written informed consent was obtained.

Adult patients over 18 years of age with a positive quick sepsis-related organ failure assessment (qSOFA) score (qSOFA score of ≥2 points) who visited the ED from July 2019 to December 2021 were initially screened by qSOFA alert system called Intelligent Sepsis Management System (i-SMS). The i-SMS system was designed as follows. Upon ED arrival, triage nurses check patient vital signs and mental status. For patients with an initial qSOFA score ≥ 2, the digital Order Communication System automatically highlights their name in violet to increase visibility to the ED physicians. Such patients are also automatically recorded into a qSOFA-positive registry group. The qSOFA alert system at our institution assists ED clinicians in identifying sepsis early and automatically enrolls patients with a positive qSOFA score [4]. A qSOFA scoring system evaluates the following three criteria and assigns one point for each criterion: altered mental status (Glasgow coma score under 15 points), high respiratory rate (≥22 breaths/min), and low blood pressure (systolic blood pressure ≤ 100 mmHg). Next, we set another inclusion criterion as an increase in the SOFA score by ≥2 points in the ED, regardless of the presence of current infection. Patients who refused to provide consent, had an increase in SOFA scores < 2, visited the hospital for trauma care, or had unknown 30-day mortality were excluded from this study. Consequently, all study participants had a qSOFA score of ≥2 points and an increase in the SOFA score by ≥2 points. Eligible participants were categorized into three groups according to the presence of infection and sepsis severity: noninfectious organ failure, sepsis, and septic shock. During the categorization process, all researchers were blinded to the levels of Gal-9, sTREM-1, and sCD25. The NIOF group was considered the control group, while the sepsis and septic shock groups were considered the experimental groups.

### 2.2. Data Collection

We collected data on demographics (sex, age, and medical history), vital signs (blood pressure, heart rate, respiratory rate, body temperature, and saturation of percutaneous oxygen [SpO_2_]), arterial blood gas analysis, laboratory results for several sepsis-related biomarkers (Gal-9, sTREM-1, sCD25, and CRP), and blood culture. Serum lactate levels were measured in all patients with sepsis according to the SSC guidelines [2]. By following up on the participants’ medical records after their ED presentation, their 30-day mortality rates were evaluated. If demographic or medical records were unavailable, clinical data were collected through telephone counseling with the patients or their legal representatives.

### 2.3. Definitions

Sepsis is now defined as a life-threatening condition marked by organ dysfunction due to a dysregulated immune response to infection, and septic shock is a more severe condition of sepsis characterized by profound cellular and metabolic abnormalities that greatly increase the risk of death [1]. Similar to our previous study [4], the criteria for NIOF included a positive qSOFA score and an increase in the SOFA score of ≥2 without the presence of current infection. The criteria for sepsis include a positive qSOFA score and an increase in SOFA score of ≥2 caused by the presence of current infection. The diagnostic criteria for septic shock include the use of vasopressors to maintain a mean arterial pressure of 65 mmHg and a serum lactate level above 2 mmol/L despite adequate fluid resuscitation. The presence of current infection was evaluated by reviewing medical records and laboratory and radiological results. The clinical severity of sepsis and septic shock was evaluated using the Acute Physiology and Chronic Health Evaluation II (APACHE II) and SOFA scores.

### 2.4. Multiplex Immunoassay

We obtained peripheral venous blood for measuring Gal-9, sTREM-1, and sCD25 levels within 6 h of ED presentation. Serum was collected in the SST-II vacutainer and stored in the Biobank of our institution at −80 °C till performing analysis. Gal-9, sTREM-1, and sCD25 levels were measured using a multiplex immunoassay based on Luminex technology (xMAP, Luminex, Austin, TX, USA). All procedures strictly followed the Luminex Assay Human Premixed Multi-Analyte Kit protocol. Heterophilic immunoglobulins were preferentially absorbed from all samples using a heteroblock (Omega Biologicals, Bozeman, MT, USA). The samples were incubated with antibody-conjugated MagPlex microspheres for 1 h with continuous shaking at room temperature, biotinylated antibodies for 1 h, and phycoerythrin-conjugated streptavidin diluted in high-quality ELISA buffer (HPE, Sanquin, Amsterdam, The Netherlands) for 10 min.

### 2.5. Statistical Analysis

Statistical analyses were performed using the SPSS (version 26.0; IBM, Armonk, NY, USA) and MedCalc (version 19.1.6; MedCalc Software, Mariakerke, Belgium) for Windows. Comparisons of continuous variables between the two groups were performed using the *t*-test or Mann–Whitney U test, according to the data distribution. Kolmogorov–Smirnov and Shapiro–Wilk tests were used to test for data normality. Comparisons of continuous variables among the three groups were performed using the Kruskal–Wallis test, which are represented as the median with an interquartile range (IQR). Statistical significance was accepted for *p*-value < 0.05, and the Bonferroni method was used to adjust *p*-value for post hoc analysis. Categorical variables were analyzed using the chi-squared test or Fisher’s exact test. Receiver operating characteristic (ROC) curve analysis was used to evaluate the diagnostic and prognostic values of the biomarkers for sepsis and septic shock. Youden’s index was used to calculate the optimal cut-off value by balancing sensitivity and specificity in ROC curve analysis. Correlations between Gal-9, sTREM-1, sCD25, and SOFA scores were analyzed using Spearman’s rank test. The prognostic value of biomarkers was evaluated using the Kaplan–Meier survival curve and Cox proportional hazard model analyses. Survival curves for 30 days stratified by the cutoff values of biomarkers were evaluated using Kaplan–Meier curve analysis and the log-rank test. A multivariable Cox proportional hazards model analysis was conducted to identify the risk factors for 30-day mortality in patients with sepsis and septic shock.

A logistic regression equation was constructed to predict the 30-day mortality probability. For this, we included biomarkers with a univariate significance of *p* < 0.1 as covariates, and 30-day mortality as the dependent variable using the backward elimination method. A logistic regression equation for predicting a logit transformation (logit (*p*)) of the probability of 30-day mortality was created using the coefficients generated for each biomarker in the final step of the regression model. Finally, the Logit (*p*) value was converted to 30-day mortality probability. The Hosmer–Lemeshow goodness-of-fit test was used to evaluate the fidelity of the regression model.

## 3. Results

### 3.1. Flowchart and Baseline Characteristics

A flowchart of the study population is shown in Figure 1. Initially, we screened 962 patients who met the positive qSOFA criteria for ED presentation using the i-SMS. Among them, 176 patients were excluded for the following reasons: refusal to provide consent (*n* = 96), increase in SOFA score < 2 (*n* = 58), ED visit for trauma care (*n* = 15), or unknown outcomes (*n* = 7). Finally, 786 patients were enrolled and classified into three groups: (1) NIOF, (2) sepsis, and (3) septic shock.

Table 1 and Figure 2 summarize the baseline characteristics of the study population. Patients with sepsis or septic shock were older than those with NIOF. The Charlson Comorbidity Index was higher for sepsis and septic shock than for NIOF. Vasopressors were administered more frequently to patients with septic shock than to those with NIOF or sepsis. SOFA and APACHE II scores and lactate levels were higher in septic shock than in NIOF or sepsis. SOFA score was higher in sepsis than in NIOF, while there was no significant difference in APACHE II score between NIOF and sepsis. The lactate levels were higher in the NIOF group than in the sepsis group. Serum levels of Gal-9, sTREM-1, sCD25, and CRP (*p* < 0.001) were higher in sepsis than in those with NIOF. Serum levels of Gal-9, sTREM-1, and sCD25 were higher in patients with septic shock than in those with sepsis. However, there was no significant difference in CRP levels between patients with sepsis and those with septic shock.

### 3.2. Correlation with Biomarkers and Severity Scores

Gal-9, sTREM-1, and sCD25 levels positively correlated with CRP (Gal-9, rho = 0.319, *p* < 0.001; sTREM-1, rho = 0.415, *p* < 0.001; sCD25, rho = 0.608, *p* < 0.001; respectively), SOFA score (Gal-9, rho = 0.351, *p* < 0.001; sTREM-1, rho = 0.454, *p* < 0.001; sCD25, rho = 0.346, *p* < 0.001; respectively), and APACEH II score (Gal-9, rho = 0.274, *p* < 0.001; sTREM-1, rho = 0.331, *p* < 0.001; sCD25, rho = 0.222, *p* < 0.001; respectively). Gal-9 and sTREM-1 correlated with lactate levels (Gal-9, rho = 0.120, *p* = 0.001; sTREM-1, rho = 0.146, *p* < 0.001, respectively), but sCD25 did not.

### 3.3. Diagnostic Value of Gal-9, sTREM-1, and sCD25

The ROC curve analyses for discriminating sepsis from NIOF and septic shock from sepsis are shown in Figure 3a,b, and Table 2 present the detailed optimal cutoff value, sensitivity, and specificity for discriminating sepsis from NIOF and septic shock from sepsis.

The optimal cut-off value of Gal-9 for discriminating sepsis from NIOF was 9719.78 ng/L (AUC, 0.638; 95% confidence interval [CI], 0.599–0.678; sensitivity, 76.3%; specificity, 47.4%; *p* < 0.001) and that for discriminating septic shock from sepsis 15,735.11 ng/L (AUC, 0.614; 95% CI, 0.562–0.667; sensitivity, 57.1%; specificity, 62.0%; *p* < 0.001). The optimal cut-off value of the sTREM-1 for discriminating sepsis from NIOF was 327.19 ng/L (AUC, 0.655; 95% CI, 0.616–0.695; sensitivity, 82.0%; specificity, 46.2%; *p* < 0.001), and that for discriminating septic shock from sepsis was 555.93 ng/L (AUC, 0.624; 95% CI, 0.572–0.676; sensitivity, 64.0; specificity, 56.0; *p* < 0.001), respectively. The optimal cut-off value of the sCD25 for discriminating sepsis from NIOF was 910.11 ng/L (AUC, 0.746; 95% CI, 0.710–0.781; sensitivity, 74.3%; specificity, 66.5%; *p* < 0.001), and that for discriminating septic shock from sepsis was 1560.53 ng/L (AUC, 0.607; 95% CI, 0.555–0.660; sensitivity, 56.6%; specificity, 63.5%; *p* < 0.001), respectively. The optimal cut-off value of the CRP for discriminating sepsis from NIOF was 36.95 mg/L (AUC, 0.843; 95% CI, 0.814–0.873; sensitivity, 81.1%; specificity, 78.9%; *p* < 0.001), and that for discriminating septic shock from sepsis was 227.35 mg/L (AUC, 0.559; 95% CI, 0.505–0.613; sensitivity, 23.3%; specificity, 89.1%; *p* = 0.032), respectively. The optimal cut-off value of the lactate for discriminating septic shock from sepsis was 2.10 mmol/L (AUC, 0.751; 95% CI, 0.707–0.795; sensitivity, 91.5%; specificity, 50.4%; *p* < 0.001).

The AUC to discriminate sepsis from NIOF was highest for CRP (AUC, 0.843), followed by sCD25 (AUC, 0.746), sTREM-1 (AUC, 0.655), Gal-9 (AUC, 0.638), and lactate (AUC, 0.519). The AUC to discriminate septic shock from sepsis was highest for lactate (AUC, 0.751), followed by sTREM-1 (AUC, 0.624), Gal-9 (AUC, 0.614), sCD25 (AUC, 0.607), and CRP (AUC, 0.559).

### 3.4. Prognostic Value of Gal-9, sTREM-1, and sCD25

ROC curve analyses using Gal-9, sTREM-1, and sCD25 levels to predict 30-day mortality are presented for the sepsis and septic shock groups (Figure 3c). The optimal cut-off values to predict 30-day mortality were 14,391.80 ng/L for Gal-9 (AUC, 0.642; 95% CI, 0.585–0.698; sensitivity, 70%; specificity, 52.9%; *p* < 0.001), 580.62 ng/L for sTREM-1 (AUC, 0.645; 95% CI, 0.589–0.701; sensitivity, 69.8%; specificity, 58.4%; *p* < 0.001), 1639.29 ng/L for sCD25 (AUC, 0.626; 95% CI, 0.567–0.685; sensitivity, 60.3%; specificity, 63.5%; *p* < 0.001), 49.5 mg/L for CRP (AUC, 0.545; 95% CI, 0.487–0.602; sensitivity, 84.9%; specificity, 28.6%; *p* = 0.141), and 4.05 mmol/L for lactate (AUC, 0.710; 95% CI, 0.657–0.764; sensitivity, 58.7%; specificity, 72.6%; *p* < 0.001), respectively.

The AUC to predict 30-day mortality was highest for lactate (AUC, 0.710), followed by sTREM-1 (AUC, 0.645), Gal-9 (AUC, 0.642), sCD25 (AUC, 0.626), and CRP (AUC, 0.545).

A multivariable logistic regression model was constructed to predict the 30-day mortality rate using biomarkers (Gal-9, sTREM-1, sCD25, CRP, and lactate) and SOFA scores and (Figure 3d). Using the regression equation, the log of the probability was converted to the probability of 30-day mortality. In the ROC curve analysis, the AUC of the combination of 3 biomarkers (Gal-9, sTREM-1, sCD25) was 0.670 (95% confidence interval [CI], 0.615–0.726; *p* < 0.001), and the model was well calibrated (Hosmer-Lemeshow test; *Χ*^2^ = 5.530; d*f* = 8; *p* = 0.700) for predicting 30-day mortality in patients with sepsis and septic shock. The AUC of the combination of 5 biomarkers (Gal-9, sTREM-1, sCD25, CRP, lactate) was 0.744 (95% CI, 0.695–0.794; *p* < 0.001), and the model was well calibrated (Hosmer-Lemeshow test, *Χ*^2^ = 14.805; d*f* = 8; *p* = 0.063). The AUC of the combination of the SOFA score and the five biomarkers (Gal-9, sTREM-1, sCD25, CRP, and lactate) was 0.740 (95% CI, 0.689–0.792, *p* < 0.001), and model was well calibrated (Hosmer-Lemeshow test, *Χ*^2^ = 10.977; d*f* = 8; *p* = 0.203).

Figure 4 shows the Kaplan–Meier survival curves stratified by the cutoff values for the probability of 30-day mortality. In all of the biomarkers tested, patients with biomarker levels over the cut-off value showed higher mortality than those with biomarker levels below the cut-off value (Gal-9 ≥ 14,391.80 ng/L, 35.9% vs. 18.0%, *p* < 0.001; sTREM-1 ≥ 580.62 ng/L, 38.5% vs. 17.0%, *p* < 0.001; sCD25 ≥ 1639.29 ng/L, 38.1% vs. 19.7%, *p* < 0.001; CRP ≥ 49.5 mg/L, 31.1% vs. 16.8%, *p* = 0.004; lactate ≥ 4.05 mmol/L, 44.3% vs. 18.2%, *p* < 0.001; respectively) (log-rank test).

Using Gal-9, sTREM-1, sCD25, CRP, and lactate, multivariable Cox proportional hazards model analysis was performed to identify the risk factors for 30-day mortality among the overall study population, including NIOF, sepsis, and septic shock (Table 3). In the overall patients, sCD25 (hazard ratio [HR], 1.000; 95% confidence interval [CI], 1.000–1.000; *p* < 0.001), CRP (HR, 1.020; 95% CI, 1.008–1.033; *p* = 0.001), and lactate levels (HR, 1.126; 95% CI, 1.100–1.154; *p* < 0.001) were identified as significant risk factors for 30-day mortality. In patients with NIOF, sTREM-1 (HR, 1.001; 95% CI, 1.000–1.001; *p* = 0.013), sCD25 (HR, 1.000; 95% CI, 1.000–1.000; *p* = 0.046), and lactate (HR, 1.107; 95% CI, 1.069–1.146; *p* < 0.001) were determined as significant risk factors for 30-day mortality. In patients with sepsis and septic shock, sCD25 (HR, 1.000; 95% CI, 1.000–1.000; *p* = 0.040) and lactate levels (HR, 1.178; 95% CI, 1.131–1.227; *p* < 0.001) were significant risk factors for 30-day mortality.

## 4. Discussion

To our knowledge, this is the largest prospective observational study on the diagnostic and prognostic value of Gal-9, sTREM-1, and sCD25 in critically ill patients with organ failure. Furthermore, this is the first study on the clinical value of Gal-9 in patients diagnosed with sepsis in the ED. Our study showed that Gal-9, sTREM-1, and sCD25 can help discriminate sepsis from NIOF, and septic shock from sepsis. These three biomarkers also have a prognostic value in patients with sepsis. The combination of biomarkers and SOFA scores showed improved performance in predicting the 30-day mortality. sCD25 is an independent risk factor for 30-day mortality in patients with sepsis.

CRP and PCT levels are widely measured in various clinical settings. CRP provides useful information on a wide range of cardiovascular events and inflammatory conditions, including sepsis, and has analytical advantages as a “robust biomarker” that is minimally affected by sample or environmental conditions [24]. However, CRP has limited value in discriminating bacterial infections and predicting severity of sepsis [25,26,27]. Although PCT is relatively specific for bacterial infections, it has a limited prognostic value in patients with sepsis. Serum lactate levels are used to detect septic shock [1]. Furthermore, it has a better prognostic value than qSOFA in patients with sepsis [28]. Several studies have demonstrated the prognostic value of lactate levels in patients with sepsis. However, lactate levels have limited value in discriminating sepsis from noninfectious diseases. Owing to the limitations of the established biomarkers, novel biomarkers with better performance are required. This study investigated Gal-9, sTREM-1, and sCD25 as potential auxiliary biomarkers of sepsis and septic shock.

Gal-9 is released from various organs during an immunologic crisis [8]. Gal-9 reflects the status of organ dysfunction; however, its clinical value has never been assessed in patients with sepsis or septic shock. In the present study, Gal-9 discriminated between sepsis and NIOF. It can also distinguish between septic shock and sepsis. However, although Gal-9 demonstrated the ability to distinguish sepsis or septic shock, its relatively low specificity should be taken into consideration, as it may increase the risk of false-positive findings. Our study showed that Gal-9 could discriminate between sepsis severities. Furthermore, the 30-day mortality differed between the two sepsis groups stratified by the cut-off value. Although Gal-9 was not superior to CRP in discriminating sepsis from NIOF, it performed better than CPR in discriminating septic shock from sepsis. Previous experimental animal studies have shown that Gal-9 has therapeutic effects in sepsis and sepsis-like models [29,30]. Moreover, T cell immunoglobulin and mucin domain 3 (Tim-3) on the T helper 1 cell surface are closely correlated with Gal-9 and form the Tim-3/Gal-9 signaling cascade [31,32]. Similarly, the binding of Gal-9 to Tim-3 showed protective effects in CD4 T cells against HIV infection [33]. Thus, Gal-9 may be a suitable auxiliary biomarker for identifying septic shock. As Gal-9 works as a powerful therapeutic mediator in the immune cascade to alleviate disease severity at the same time, elevated levels of Gal-9 alone in septic conditions should not be simply interpreted as the pathological severity of sepsis.

sTREM-1 could discriminate sepsis severity, and the 30-day survival curves differed between the two sepsis groups stratified by their cutoff values. sTREM-1 levels increase in the early phase of sepsis and decrease after adequate treatment [34]. Another study showed that sTREM-1 could effectively predict 28-day mortality in patients with sepsis, severe sepsis, and septic shock [16]. However, in that study, the prognostic value of sTREM-1 was not superior to that of the SOFA and APACHE II scores. sTREM-1 did not show better performance as a diagnostic biomarker for severe sepsis and septic shock compared to CRP and IL-6 [35]. Another prospective observational study suggested that sTREM-1 is a better predictor of 90-day mortality than CRP and PCT in septic shock [36]. These discrepancies might be partly caused by the different disease severities of the study populations or the different control group settings. A prospective cohort study suggested that sTREM-1 levels in patients with sepsis admitted to the ICUs could reflect infectious conditions more accurately than CRP and PCT levels [37]. Despite these controversial results, sTREM-1 appears to have significant diagnostic and prognostic value in sepsis. However, sTREM-1 also had relatively low specificity for discriminating sepsis or septic shock in the present study. Therefore, its potential risk of false-positive findings should be taken into consideration.

Several studies have assessed the clinical value of sCD25 as a sepsis-related biomarker. A previous study on serum protein markers suggested sCD25 as a complementary tool for diagnosing sepsis [38]. Increased plasma sCD25 levels and Treg percentages are associated with sepsis [39]. Similar to these studies, our study showed that sCD25 could discriminate sepsis from NIOF, and septic shock from sepsis. Although sCD25 was superior to sTREM-1 and sCD25 in discriminating sepsis from NIOF, it was not superior to either sTREM-1 or sCD25 in discriminating septic shock from sepsis. This may be explained by decreased sCD25 levels before death caused by sepsis, which reflects immune suppression and exhaustion of activated T-lymphocytes [18]. Among Gal-9, sTREM-1, and sCD25, only sCD25 was found to be an independent risk factor for 30-day mortality in patients with sepsis in the Cox proportional hazards model. Although sCD25 levels did not correlate with disease severity, sCD25 could effectively predict mortality in patients with sepsis in ICUs [40]. Another study suggested sIL-2Rα (i.e., sCD25) for the prediction of sepsis occurrence in multiple trauma patients [41]. Overall, sCD25 appears to have both diagnostic and prognostic value in patients with sepsis and septic shock. Furthermore, our study provides supportive evidence for the robust prognostic value of sCD25 levels.

Although Gal-9, sTREM-1, and sCD25 demonstrated statistically significant clinical value in patients with sepsis, their AUC values were not higher than those of established biomarkers such as CRP and lactate. This suggests that, in routine clinical practice, these biomarkers may be useful as adjunctive tools rather than standalone indicators. Alternatively, as we proposed, the development of a multivariable model including the conventional biomarkers could potentially improve prognostic performance. Further studies are required to determine whether they provide superior diagnostic or prognostic value in specific diseases.

PCT has been widely used to diagnose and prognosticate sepsis patients. However, as PCT levels were not measured in approximately 15% of our study population, we did not include PCT levels along with Gal-9, sTREM-1, and sCD25 in the main results. To compare the clinical value of PCT with that of other markers, we additionally performed sensitivity analysis including only patients who had procalcitonin measurements (Appendix A). In this analysis, the AUC values of procalcitonin were 0.784 (95% CI, 0.746–0.822; *p* < 0.001) for discriminating sepsis from NIOF, 0.691 (95% CI, 0.641–0.741; *p* < 0.001) for discriminating septic shock from sepsis, and 0.539 (95% CI, 0.481–0.597; *p* = 0.204) for predicting 30-day mortality. The results for Gal-9, sTREM-1, and sCD25 were similar to the main analyses. PCT showed better performance in discriminating sepsis or septic shock compared with Gal-9, sTREM-1, and sCD25, but it could not effectively predict 30-day mortality compared with the 3 tested biomarkers. In accordance with our results, previous studies reported that PCT had good diagnostic performance but limited prognostic value in patients with sepsis [42,43].

There are several limitations in the current study. First, this was a single-center, ED-based study, which has limited external validity. Therefore, further multi-center, ICUs- or other EDs-based studies are recommended to support our results. Second, only the initial levels of individual biomarkers were measured in the ED, and subsequent changes were not determined. Dynamic monitoring of biomarkers can help diagnose and prognosticate patients with sepsis; thus, further studies, including follow-up changes in these markers, are needed. Third, because our study included only patients with organ failure screened in the ED, this might have resulted in a selection bias. Because we focused primarily on the ability of the tested biomarkers to discriminate the presence of infection among critically ill patients, we selected the NIOF group—patients without infection within the context of sepsis (i.e., infection with organ failure)—as the control. Although NIOF was used as a control group, there may have been patients who had an infection but no organ failure (infection without organ failure) during the study period. There may also have been patients without infection or organ failure. Because we did not enroll these patients, our results may have underestimated the discriminating power of the biomarkers. To determine the generalizability of our findings, studies including control groups of different severity or characteristics are required. Fourth, this study analyzed only single measurements of tested biomarkers. Because serial measurements can provide greater prognostic value than single measurements, future studies with serial biomarker analysis are required.

## 5. Conclusions

Gal-9, sTREM-1, and sCD25 have diagnostic and prognostic value in critically ill patients with organ failure. Among these, sCD25 showed the best performance in distinguishing sepsis from NIOF. Furthermore, sCD25 level was an independent risk factor for 30-day mortality among patients with sepsis. Overall, Gal-9, sTREM-1, and sCD25 could be used as potential auxiliary biomarkers for supporting clinical decision in critically ill patients, including those with sepsis and septic shock. To confirm the generalizability of our findings, multicenter prospective studies including different control groups and serial biomarker measurements are required.

## Figures and Tables

**Figure 1 diagnostics-15-02677-f001:**
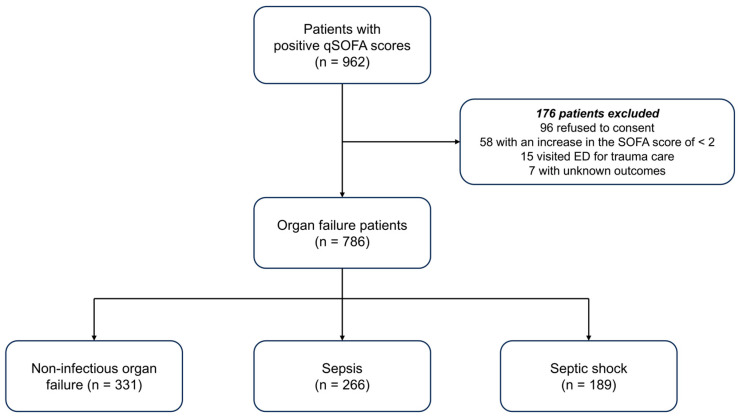
Flowchart of the study population. qSOFA, quick sepsis-related organ failure assessment; SOFA, sepsis-related organ failure assessment; ED, emergency department.

**Figure 2 diagnostics-15-02677-f002:**
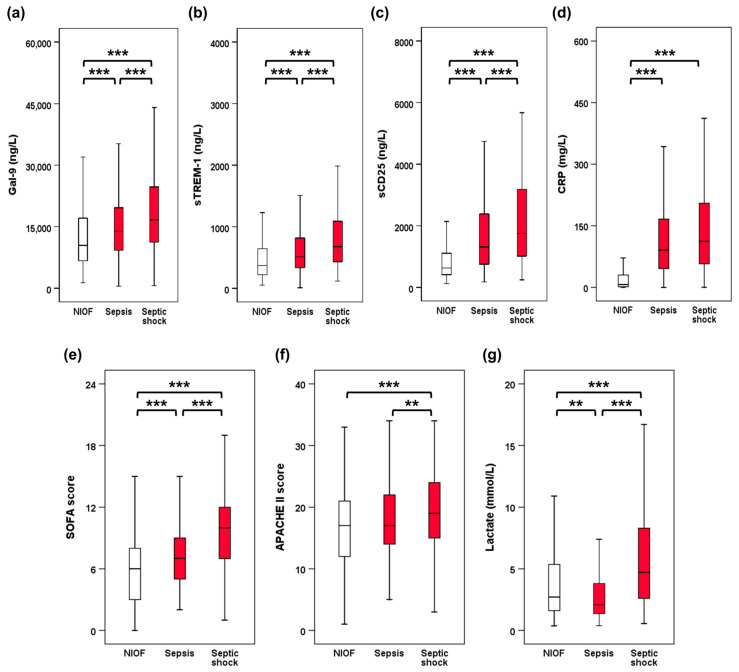
Biomarker levels and clinical severity scores between 3 groups. (**a**) Gal-9. (**b**) sTREM-1. (**c**) sCD25. (**d**) CRP. (**e**) SOFA score. (**f**) APACHE II score. (**g**) Lactate. ** *p* < 0.01, *** *p* < 0.001. Gal-9, galectin-9; sTREM-1, soluble triggering receptor expressed on myeloid cells-1; sCD25, soluble CD25; CRP, C-reactive protein; SOFA, sepsis-related organ failure assessment; APACHE II, Acute Physiology and Chronic Health Evaluation II.

**Figure 3 diagnostics-15-02677-f003:**
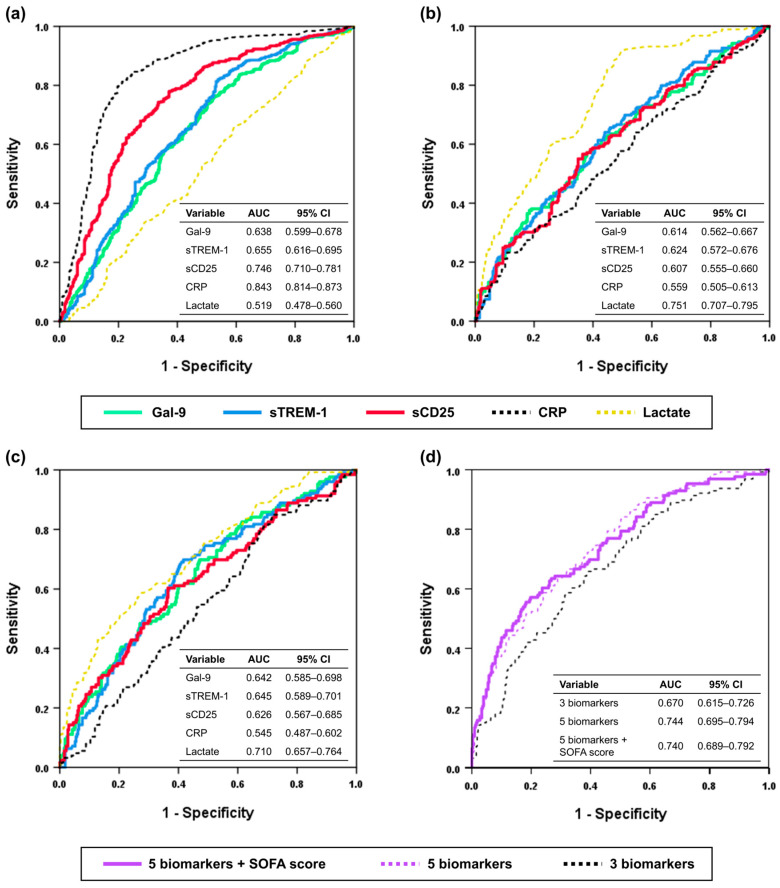
ROC curve analyses. (**a**) Discriminating sepsis from NIOF. (**b**) Discriminating septic shock from sepsis. (**c**) Predicting 30-day mortality in sepsis including septic shock. (**d**) combination of SOFA score and biomarkers for predicting 30-day mortality in sepsis including septic shock. Three biomarkers include Gal-9, sTREM-1, and sCD25, and five biomarkers include Gal-9, sTREM-1, sCD25, CRP, and lactate. ROC, Receiver operating characteristic; Gal-9, galectin-9; sTREM-1, soluble triggering receptor expressed on myeloid cells-1; sCD25, soluble CD25; CRP, C-reactive protein; NIOF, non-infectious organ failure.

**Figure 4 diagnostics-15-02677-f004:**
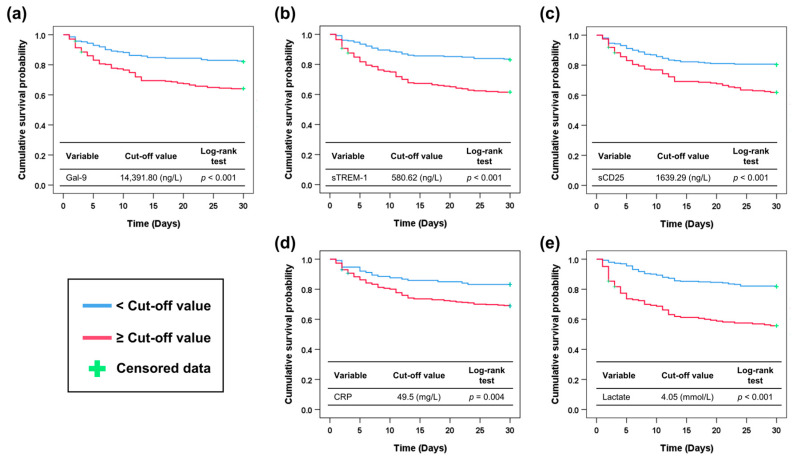
Kaplan–Meier survival curve according to the optimal cut-off values of biomarkers to predict 30-day mortality. (**a**) Gal-9. (**b**) sTREM-1. (**c**) sCD25. (**d**) CRP. (**e**) Lactate. Gal-9, galectin-9; sTREM-1, soluble triggering receptor expressed on myeloid cells-1; sCD25, soluble CD25; CRP, C-reactive protein.

**Table 1 diagnostics-15-02677-t001:** Baseline characteristics of the study population.

Variable	NIOF (*n* = 331)	Sepsis (*n* = 266)	Septic Shock (*n* = 189)	*p*-Value
Age, median (IQR)	67 (51–82)	77 (69–84)	79 (68–84)	<0.001
Male, *n* (%)	181 (54.7)	154 (57.9)	109 (57.7)	0.684
Charlson’s morbidity index, median (IQR)	5 (3–7)	6 (5–8)	6 (5–8)	<0.001
Past medical history, *n* (%)				
Myocardial infarction	11 (3.3)	7 (2.6)	1 (0.5)	0.131
Chronic heart disease	28 (8.5)	25 (9.4)	17 (9.0)	0.922
Peripheral vascular disease	161 (48.6)	138 (51.9)	107 (56.6)	0.215
Cerebrovascular disease	74 (22.4)	101 (38.0)	78 (41.3)	<0.001
Dementia	42 (12.7)	57 (21.4)	51 (27.0)	<0.001
COPD	27 (8.2)	34 (12.8)	17 (9.0)	0.152
Diabetes	90 (27.2)	106 (39.8)	66 (34.9)	0.004
Connective tissue disease	3 (0.9)	7 (2.6)	6 (3.2)	0.148
Peptic ulcer disease	18 (5.4)	13 (4.9)	6 (3.2)	0.496
Liver disease	38 (11.5)	9 (3.4)	13 (6.9)	0.001
Hemiplegia	27 (8.2)	68 (25.6)	55 (29.1)	<0.001
Chronic kidney disease (stage ≥ 3)	31 (9.4)	20 (7.5)	12 (6.3)	0.445
Malignancy	73 (22.1)	57 (21.4)	38 (20.1)	0.873
Vital signs, median (IQR)				
Systolic blood pressure (mmHg)	100 (91–145)	104 (91–139)	91 (80–115)	<0.001
Diastolic blood pressure (mmHg)	64 (53–85)	63 (54–79)	55 (48–69)	<0.001
Heart rate (rate/min)	98 (80–119)	108 (89–124)	110 (88–128)	<0.001
Respiratory rate (breath/min)	24 (22–26)	24 (20–28)	24 (20–30)	0.265
Body temperature (°C)	36.5 (36.0–37.0)	37.2 (36.4–38.2)	36.9 (36.1–38.0)	<0.001
SpO_2_ (%)	97 (93–99)	96 (93–99)	93 (86–97)	<0.001
Vasopressor administration, *n* (%)	53 (16.0)	32 (12.0)	121 (64.0)	<0.001
Platelet (×10^9^/L), median (IQR)	209 (157–289)	206 (138–287)	186 (117–245)	0.001
Bilirubin (mg/L), median (IQR)	5.60 (3.10–10.80)	6.50 (4.00–10.80)	6.90 (4.90–12.90)	0.002
Creatinine (mg/L), median (IQR)	11.20 (8.00–19.30)	12.30 (8.00–20.40)	15.70 (10.40–24.00)	<0.001
WBC (×10^9^/L), median (IQR)	10.9 (8.0–15.5)	12.2 (8.2–18.7)	11.0 (6.6–16.9)	0.016
Length of hospital stay (days), median (IQR)	12 (6–20)	13 (8–23)	16 (9–31)	0.002

NIOF, noninfectious organ failure; IQR, interquartile range; COPD, chronic obstructive pulmonary disease; SpO_2_, saturation of percutaneous oxygen; WBC, white blood cell.

**Table 2 diagnostics-15-02677-t002:** Discriminating powers of the biomarkers presented as areas under the curve (95% CI).

Biomarker	AUC (95% CI)	*p*-Value	Cut-Off Value	Sensitivity	Specificity
Gal-9					
NIOF vs. * Sepsis	0.638 (0.599–0.678)	<0.001	9719.78 (ng/L)	76.3%	47.4%
Sepsis vs. Septic shock	0.614 (0.562–0.667)	<0.001	15,735.11 (ng/L)	57.1%	62.0%
sTREM-1					
NIOF vs. * Sepsis	0.655 (0.616–0.695)	<0.001	327.19 (ng/L)	82.0%	46.2%
Sepsis vs. Septic shock	0.624 (0.572–0.676)	<0.001	555.93 (ng/L)	64.0%	56.0%
sCD25					
NIOF vs. * Sepsis	0.746 (0.710–0.781)	<0.001	910.11 (ng/L)	74.3%	66.5%
Sepsis vs. Septic shock	0.607 (0.555–0.660)	<0.001	1560.53 (ng/L)	56.6%	63.5%
CRP					
NIOF vs. * Sepsis	0.843 (0.814–0.873)	<0.001	36.95 (mg/L)	81.1%	78.9%
Sepsis vs. Septic shock	0.559 (0.505–0.613)	0.032	227.35 (mg/L)	23.3%	89.1%
Lactate					
NIOF vs. * Sepsis	0.519 (0.478–0.560)	0.361	2.12 (mmol/L)	65.9%	40.8%
Sepsis vs. Septic shock	0.751 (0.707–0.795)	<0.001	2.10 (mmol/L)	91.5%	50.4%

AUC, area under the curve; Gal-9, galectin-9; sTREM-1, soluble triggering receptor expressed on myeloid cells-1; sCD25, soluble CD25; CRP, C-reactive protein; NIOF, noninfectious organ failure. * Sepsis, including septic shock.

**Table 3 diagnostics-15-02677-t003:** Multivariable Cox proportional hazards models of risk factors for 30-day mortality.

Biomarker	All Patients (*n* = 786)		NIOF (*n* = 331)		* Sepsis (*n* = 455)	
	Multivariable HR (95% CI)	*p*-Value	Multivariable HR (95% CI)	*p*-Value	Multivariable HR (95% CI)	*p*-Value
Gal-9	-	** ns	-	ns	-	ns
sTREM-1	-	ns	1.001 (1.000–1.001)	0.013	-	ns
sCD25	1.000 (1.000–1.000)	<0.001	1.000 (1.000–1.000)	0.046	1.000 (1.000–1.000)	0.040
CRP	1.020 (1.008–1.033)	0.001	-	ns	-	ns
Lactate	1.126 (1.100–1.154)	<0.001	1.107 (1.069–1.146)	<0.001	1.178 (1.131–1.227)	<0.001

HR, hazard ratio; CI, confidence interval; Gal-9, galectin-9; sTREM-1, soluble triggering receptor expressed on myeloid cells-1; sCD25, soluble CD25; CRP, C-reactive protein; NIOF, noninfectious organ failure. * Sepsis, including septic shock; ** ns, not significant.

## Data Availability

The data are not publicly available due to privacy of the study participants. The datasets used and/or analyzed during the current study are available from the corresponding author s on reasonable request.

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
