# Peer review of "Clinical Value of Galectin-9, Soluble TREM-1, and Soluble CD25 Among Critically Ill Patients with Organ Failure in the Emergency Department: A Prospective Observational Study"

_diagnostics, 2025, doi:10.3390/diagnostics15212677_

Round 1

Reviewer 1 Report

Comments and Suggestions for Authors

The study is devoted to assessing the diagnostic value of three biomarkers for dividing intensive care patients into groups of multiple organ failure without infection, sepsis, and septic shock. The purpose of the study and its design are clearly formulated, the sample in each group is impressive in size, which allows us to assess the results obtained in the study as adequate.

There are no comments on the Introduction and Discussion as a whole. However, the list of references includes publications up to and including 2022. Considering that the study itself was also conducted from 2019 to 2021, the literature data provided may indicate that the search for more relevant information for the past 3 years is not being conducted. Perhaps it should be conducted, or it should be indicated in the text that no new relevant information on this topic has been published in recent years.

The main comments can be formulated in terms of statistical processing of the data obtained and their presentation in the Results section.
1) First of all, Table 1 contains a not very clear way of presenting statistically significant differences between groups. If a pairwise comparative analysis is performed using the Mann-Whitney test with Bonferroni correction, then it is better to provide all three groups of p values. If the Kruskal-Wallis test was performed, then one p value is given.

2) The results of the ROC analysis in section 3.3. demonstrate moderate prognostic significance of all the studied parameters. At the same time, it can be noted that some parameters are highly sensitive, while others are highly specific. A logical conclusion can be a proposal to build multivariate models that could include both the 3 biomarkers studied, as well as lactate and CRP, which have good prognostic characteristics. This proposal is an urgent recommendation for the authors, since single-parameter prognostic models do not have the desired area under the curve of more than 0.9 in any of the cases, which means that the diagnostic significance of the data presented is generally quite moderate.

3) Lactate as a marker of hypoxia and acidosis, and CRP as a marker of inflammation are not the best candidates for comparing biomarkers directly in the case of sepsis diagnosis. Procalcitonin would be the best marker for comparison here. Was procalcitonin determined in your study? If so, then data on it should be added to the article and analyzed.

4) A small note: sCD24 is indicated in the last paragraph of the Introduction, but no information on it was presented in the text before that. Either this is a typo, or the information needs to be supplemented.

Author Response

Reviewer 1

There are no comments on the Introduction and Discussion as a whole. However, the list of references includes publications up to and including 2022. Considering that the study itself was also conducted from 2019 to 2021, the literature data provided may indicate that the search for more relevant information for the past 3 years is not being conducted. Perhaps it should be conducted, or it should be indicated in the text that no new relevant information on this topic has been published in recent years.

Answer: Thank you for your helpful comments. According to your advice, we conducted a recent literature review for the past 3 years and identified some studies which investigated the performance of Gal-9 for predicting the risk of malignancy in dermatomyositis, prognostic value of TREM-1 in sepsis, and diagnostic value of sCD25 for neonatal sepsis. We have cited these studies in the latter part of introduction section. Thanks to your valuable comments, we were able to improve and update our manuscript.

1) First of all, Table 1 contains a not very clear way of presenting statistically significant differences between groups. If a pairwise comparative analysis is performed using the Mann-Whitney test with Bonferroni correction, then it is better to provide all three groups of p values. If the Kruskal-Wallis test was performed, then one p value is given.

Answer: Thank you for your valuable comment. Our original intention was to present both the results of pairwise comparisons using the Mann-Whitney test with Bonferroni correction and those of the Kruskal-Wallis test. However, because this method may render the presentation of statistical significance unclear, we have revised Table 1 to show only the p-values derived from the Kruskal-Wallis test. Additionally, data on tested biomarkers and clinical severity scores were presented in separate figure (medians with IQRs) (Figure 2)

2) The results of the ROC analysis in section 3.3. demonstrate moderate prognostic significance of all the studied parameters. At the same time, it can be noted that some parameters are highly sensitive, while others are highly specific. A logical conclusion can be a proposal to build multivariate models that could include both the 3 biomarkers studied, as well as lactate and CRP, which have good prognostic characteristics. This proposal is an urgent recommendation for the authors, since single-parameter prognostic models do not have the desired area under the curve of more than 0.9 in any of the cases, which means that the diagnostic significance of the data presented is generally quite moderate.

Answer: Thank you for your valuable comments. First, the prognostic value is described in Section 3.4 rather than Section 3.3. In addition, the multivariable analysis presented in Figure 3 demonstrates the ability to predict 30-day mortality, not sepsis diagnosis. Our original analysis illustrated how the biomarkers can enhance the performance of the SOFA score for predicting 30-day mortality. However, according to your suggestion, we have reanalyzed the data focusing on the 3 biomarkers (Gal-9, sTREM-1, and sCD25), and the figure has been revised and updated. This may help readers understand how the combination of these 3 biomarkers can improve prognostic performance compared with CRP and lactate alone. Although the AUC of 5 biomarker combination using multivariable logistic regression equation did not reach 0.9, it showed greater value than that of lactate alone (0.744 vs 0.710). This multi-marker approach highlights the potential of these 3 biomarkers to serve as adjunctive prognostic tools to the established biomarkers in clinical practice.

3) Lactate as a marker of hypoxia and acidosis, and CRP as a marker of inflammation are not the best candidates for comparing biomarkers directly in the case of sepsis diagnosis. Procalcitonin would be the best marker for comparison here. Was procalcitonin determined in your study? If so, then data on it should be added to the article and analyzed.

Answer: Thank you for the valuable comments. We described this point in the limitations of the discussion section. Because our study primarily focused on the measurements of Galectin-9, sTREM-1, and sCD25, procalcitonin levels were missing in approximately 15% of the enrolled patients. Given the significant proportion of missing data, reliable statistical adjustment was not feasible. Therefore, we did not include procalcitonin as a main tested biomarker to compare with others. However, according to your comments, we performed a sensitivity analysis including only patients with procalcitonin measurements to determine the diagnostic and prognostic performance. Because of significant proportion of missing data in procalcitonin levels, it seems to be inappropriate to include procalcitonin as a tested biomarker in the main results. Therefore, we have included this result as a supplementary file in the last part of our manuscript.

4) A small note: sCD24 is indicated in the last paragraph of the Introduction, but no information on it was presented in the text before that. Either this is a typo, or the information needs to be supplemented.

Answer: Thank you for the accurate comments. Our study investigated sCD25, not sCD24. Definitely, “sCD24” was a typographical error. This erroneous mention has been removed from the manuscript.

Reviewer 2 Report

Comments and Suggestions for Authors

I read with interest paper entitled „Clinical value of galectin-9, soluble TREM-1, and soluble CD25 among critically ill patients with organ failure in the emergency department: a prospective observational study“. This is a large prospective observational study that evaluates the diagnostic and prognostic value of Galectin-9, soluble TREM-1, and soluble CD25 in patients with organ failure presenting to the emergency department. This manuscript is well written, and the authors address an interesting research question.

However, there are several issues that should be addressed to improve the paper:

  1. The added diagnostic/prognostic value over well-established biomarkers (e.g., CRP, PCT, lactate) is modest, with low AUC values. The authors should more explicitly acknowledge the limited value and discuss whether these markers could realistically be incorporated into clinical workflows.
  2. The choice of non-infectious organ failure as the control group may have introduced bias and this could overestimate diagnostic accuracy. Please elaborate on the rationale for this selection and its implications for generalizability.
  3. The absence of serial measurements is an important limitation, particularly since the trajectory of biomarkers often carries greater prognostic value than a single value. This should be highlighted more clearly in the discussion and conclusions.
  4. The practical cut-off values provided are useful. However, the relatively low specificity of Gal-9 and sTREM-1 (around 50–60%) raises concerns about false positives. The discussion should acknowledge this.
  5. Table 1 is dense. Highlighting the most relevant variables (biomarkers, scores, outcomes) would improve readability. Data on Gal-9, sTREM-1 and sCD25 can be presented in separate figure (medians with IQRs). Summarise main AUC values in the Results section instead of depending mostly on tables or figures.

Author Response

I read with interest paper entitled “Clinical value of galectin-9, soluble TREM-1, and soluble CD25 among critically ill patients with organ failure in the emergency department: a prospective observational study”. This is a large prospective observational study that evaluates the diagnostic and prognostic value of Galectin-9, soluble TREM-1, and soluble CD25 in patients with organ failure presenting to the emergency department. This manuscript is well written, and the authors address an interesting research question.

Answer: Thank you for your time and thoughtful comments to our manuscript. We authentically revised our article according to your advice.

1) The added diagnostic/prognostic value over well-established biomarkers (e.g., CRP, PCT, lactate) is modest, with low AUC values. The authors should more explicitly acknowledge the limited value and discuss whether these markers could realistically be incorporated into clinical workflows.

Answer: Thank you for your valuable comments. As you commented, the diagnostic and prognostic value of single tested biomarker is modest, with limited AUC values. To attenuate the limitation of single marker approach, we combined the tested biomarkers with established biomarkers and SOFA score. We suggest that combination of biomarkers and clinical scores could help diagnose or prognosticate sepsis patient in the clinical workflows. Despite this multi-marker approach, we explicitly acknowledge the limited value of tested biomarkers and have described this point in the Discussion section.

2) The choice of non-infectious organ failure as the control group may have introduced bias and this could overestimate diagnostic accuracy. Please elaborate on the rationale for this selection and its implications for generalizability.

Answer: Thank you for the valuable comments. The potential risk of bias associated with selecting the non-infectious organ failure (NIOF) group as the control, as well as its implications for generalizability, have been addressed in the final paragraph of the Discussion. The rationale for choosing the NIOF group as the control was that we considered the ability of the three biomarkers to distinguish the presence of infection—a primary component in sepsis diagnosis—to be of key importance among critically ill patients. Therefore, we selected a cohort without infection within the context of Sepsis (Infection + Organ Failure). This point has been explicitly added as a limitation of the study in the manuscript.

NIOF group includes patients with severe organ failure such as pulmonary thromboembolism, decompensated heart failure, cardiac tamponade, and status epilepticus, where infection is absent but the clinical severity is nonetheless profound. Using such critical patients as the control group is more likely to underestimate the discriminative ability of the biomarkers for diagnosing sepsis. In contrast, if normal healthy controls or mild patients were included in the control group, the discriminating value of biomarkers would be overestimated. In our previous study where the NIOF group served as the control, the performance of the biomarkers was not overestimated compared with previous studies [1].

  1. Lee, S., Song, J., Park, D. W., Seok, H., Ahn, S., Kim, J., Park, J., Cho, H. J., & Moon, S. (2022). Diagnostic and prognostic value of presepsin and procalcitonin in non-infectious organ failure, sepsis, and septic shock: a prospective observational study according to the Sepsis-3 definitions. BMC infectious diseases22(1), 8. https://doi.org/10.1186/s12879-021-07012-8

3) The absence of serial measurements is an important limitation, particularly since the trajectory of biomarkers often carries greater prognostic value than a single value. This should be highlighted more clearly in the discussion and conclusions.

Answer: Thank you for the valuable comments. We totally agree with your comments that not analyzing serial measurements is an important limitation of our study. We have clearly addressed this point in the last paragraph of the Discussion and Conclusions.

4) The practical cut-off values provided are useful. However, the relatively low specificity of Gal-9 and sTREM-1 (around 50–60%) raises concerns about false positives. The discussion should acknowledge this.

Answer: Thank you for the valuable comments. The relatively low specificity of Gal-9 and sTREM-1, as well as the potential risk of false-positive findings, have been explicitly described in the 3rd and 4th paragraph of Discussion.

5) Table 1 is dense. Highlighting the most relevant variables (biomarkers, scores, outcomes) would improve readability. Data on Gal-9, sTREM-1 and sCD25 can be presented in separate figure (medians with IQRs). Summarise main AUC values in the Results section instead of depending mostly on tables or figures.

Answer: Thank you for the valuable advice. According to your recommendations, data on Gal-9, sTREM-1 and sCD25 were presented in separate figure (medians with IQRs) (Figure 2). In additions, we have summarized main AUC values (diagnostic and prognostic) in the Results (Sections 3.3 and 3.4).

Round 2

Reviewer 1 Report

Comments and Suggestions for Authors

I thank the authors for their detailed responses. The authors performed the suggested additional statistical analysis of the data. Furthermore, at the end of the Q&A, the authors provided additional data, including data on procalcitonin, but I did not see any mention of this data in the manuscript. I suppose that including this data as supporting material in the article, along with the limited number of patients in whom it was measured, is appropriate and useful for comparison. Furthermore, in the discussion section, it would be interesting to see how the authors explain the low prognostic value of procalcitonin, as the gold standard for diagnosing sepsis, in their groups. Perhaps this could also explain the less-than-ideal prognostic value of other parameters.

Author Response

Comments 1: I thank the authors for their detailed responses. The authors performed the suggested additional statistical analysis of the data. Furthermore, at the end of the Q&A, the authors provided additional data, including data on procalcitonin, but I did not see any mention of this data in the manuscript. I suppose that including this data as supporting material in the article, along with the limited number of patients in whom it was measured, is appropriate and useful for comparison. Furthermore, in the discussion section, it would be interesting to see how the authors explain the low prognostic value of procalcitonin, as the gold standard for diagnosing sepsis, in their groups. Perhaps this could also explain the less-than-ideal prognostic value of other parameters.

Response 1: Thank you for your thoughtful comments and recommendations. We have provided the analysis including procalcitonin levels as a supplementary figure, and presented the results in the Discussion section (7th paragraph) to help readers understand our findings better. Furthermore, according to your advice, we have mentioned the good diagnostic performance and limited prognostic value of procalcitonin in the discussion section. We really appreciate your accurate comments.

Reviewer 2 Report

Comments and Suggestions for Authors

I do not have any further complaints.

Author Response

Comments 1: I do not have any further complaints.

Response 1: We really appreciate your thoughtful comments. Thanks to your advice, we have improved our manuscript.